# Non-microbial methane emissions from tropical rainforest soils under different conditions

Gaohui Jia[1], Qiu Yang[1]*, Huai Yang[2], Yamin Jiang[1], Wenjie Liu[1,3]*, Tingting Wu[4], Han Mao[1], Tianyan Su[1], Zhenghong Tan[1], Xu Wang[1], Juelei Li[5]

**1** Key Laboratory of Agro-Forestry Environmental Processes and Ecological Regulation of Hainan Province, College of Ecology and Environment, Hainan University, Haikou, Hainan, China, **2** International Center for Bamboo and Rattan, Beijing, China, **3** Center for Ecosystem Science and Society, Northern Arizona University, Flagstaff, Arizona, United States of America, **4** School of Ecology, Sun Yat-sen University, Guangzhou, Guangdong, China, **5** School of Science and Ocean science, Hong Kong University of Science and Technology, Hong Kong, China

* yangqiu0903@163.com (QY); liuwj@hainanu.edu.cn (WL)

**Data Availability Statement:** All relevant data are within the manuscript.

**Funding:** This work was supported by the Natural Science Foundation of Hainan province (No. 418MS019 and 2019RC012); the National Key R &

## Abstract

Non-microbial methane (NM-$CH_4$), emissions from soil might play a significant role in carbon cycling and global climate change. However, the production mechanisms and emission potential of soil NM-$CH_4$ from tropical rainforest remain highly uncertain. In order to explore the laws and characteristics of NM-$CH_4$ emission from tropical rainforest soils. Incubation experiments at different environmental conditions (temperatures, soil water contents, hydrogen peroxide) and for soils with different soil organic carbon (SOC) contents were conducted to investigate the NM-$CH_4$ emission characteristics and its influence factors of soils (0-10cm) that collected from a tropical rainforest in Hainan, China. Incubation results illustrated that soil NM-$CH_4$ release showed a linear increase with the incubation time in the first 24 hours at 70 ˚C, whereas the logarithmic curve increase was found in 192 h incubation. Soil NM-$CH_4$ emission rates under aerobic condition were significantly higher than that of under anaerobic condition at first 24 h incubation. The increasing of temperature, suitable soil water contents (0–100%), and hydrogen peroxide significantly promoted soil NM-$CH_4$ emission rates at the first 24 h incubation. However, excessive soil water contents (200%) inhibited soil NM-$CH_4$ emissions. According to the curve simulated from the NM-$CH_4$ emission rates and incubation time at 70 ˚C of aerobic condition, soil would no longer release NM-$CH_4$ after 229 h incubation. The NM-$CH_4$ emissions were positively corelated with SOC contents, and the average soil NM-$CH_4$ emission potential was about 6.91 ug per gram organic carbon in the tropical mountain rainforest. This study revealed that soils in the tropical rainforest could produce NM-$CH_4$ under certain environment conditions and it supported production mechanisms of thermal degradation and reactive oxygen species oxidation. Those results could provide a basic data for understanding the soil NM-$CH_4$ production mechanisms and its potential in the tropical rainforest.

D Program of China (NO. 2018YFD0201105); the National Natural Science Foundation of China (No. 41663010). The author of Wenjie Liu thanks for the financial support from China Scholarship Council.

**Competing interests:** The authors have declared that no competing interests exist.

## Introduction

Methane ($CH_4$), an important greenhouse gas, which contribution to the greenhouse effect is second only to $CO_2$, and has a major impact on atmospheric chemistry and climate [1]. The concentration of $CH_4$ in the atmosphere has been increasing since the industrial revolution. It is expected that by the 2030s, the amounts of $CH_4$ released by human activities will increase by about 25% [2], which may have a major impact on climate change in the future. Nearly 15 to 30 percent of $CH_4$ in the atmosphere come from soils each year [3]. Studies focus on $CH_4$ emission mechanisms and potential from soil are critical in understanding carbon cycling and global climate change projections.

Generally, there are two production mechanisms of soils for atmospheric $CH_4$, one is microbial $CH_4$ and the other is non-microbial methane (NM-$CH_4$) [4]. Soil microbial $CH_4$, produced by the methanogenesis of methanogen, which was considered to be the mainly sources of soil $CH_4$ emissions. Hao et al. [5] first observed that $CH_4$ could be released from the soil in the savanna grassland during the dry season, and the $CH_4$ emission was also detected in the forest soil. Subsequently, Andersen et al. [6] and Fischer and Hedin [7] found soil $CH_4$ emission in aerobic soils. There are many explanations for the possible underlying mechanisms for this phenomenon, which are based on the microbial perspectives. Rimbault et al. [8] reported that the soil aerobic bacteria which could produce a small amount of $CH_4$ under certain conditions, while Peter and Conrad [9] reported that the existence of soil anaerobic microhabitats can explain this phenomenon.

Compared with microbial $CH_4$, soil NM-$CH_4$ has received less attention in the past decades. The production of soil NM-$CH_4$ is more extensive than we originally thought in the soil environment. Soil NM-$CH_4$ emission is an instantaneous reaction product of organic compounds under environmental pressures, which was caused by the cutting off the methyl functional groups of organic compounds [10]. The sources of NM-$CH_4$ mainly include energy utilization [11], biomass combustion [12], and geological release [13]. In recent years, it has been discovered that plants [14], animals [15], and marine surface water [16] can also produce NM-$CH_4$ under high temperature [17,18], strong ultraviolet radiation [19] and rich reactive oxygen [20]. Since plants were proved to release NM-$CH_4$, some research scholars wonder whether soil could also release NM-$CH_4$ [21]. This new perspective was first demonstrated by Kammann et al. [22], who found that soil samples can still release large amounts of $CH_4$ even after homogenization (the anaerobic habitats in oil have been destroyed). Then, several recent studies have also confirmed that soil can produce NM-$CH_4$ under aerobic conditions [10,23–25].

Although some researches on soil NM-$CH_4$ emissions and its influencing factors were reported in recent years [22–25], the production mechanisms and emission potential of the NM-$CH_4$ from tropical rainforest soil remain highly uncertain because of limited study. Tropical rainforests, as one of the most important components of forest, are of great importance for the global carbon cycle. To study NM-$CH_4$ emissions of tropical rainforest soils is of great importance in understanding the carbon cycle of forest ecosystems and $CH_4$ emission reducing. The Jianfengling Long-term Research Station of Tropical Forest Ecosystem is an unique platform to address the release mechanisms and potential of soil NM-$CH_4$ emissions. Soil samples at depth of 0–10 cm were collected from 6 plots (20 m × 20 m) in Jianfengling Long-term Research Station of Tropical Forest Ecosystem, and the NM-$CH_4$ emission rates from soil incubation experiments were observed under four different incubation conditions. The aims of this study are to understand the possible releasing mechanisms and potential of soil NM-$CH_4$ emissions in tropical rainforest. In this study we hypothesized that NM-$CH_4$ releases rates would decrease with incubation time, and the soil NM-$CH_4$ could be influenced by

different temperature, water content condition, hydrogen peroxide ($H_2O_2$) and soil organic carbon contents.

## Materials and methods

### Study area

The study area located in Jianfengling National Nature Reserve (18.33˚~18.95˚N, 108.48˚~ 109.20˚E, altitude range: 0-1412m), south-west Hainan Island, China. This area is characterized by a tropical rainforest climate with a mean annual precipitation of approximately 2400 mm (of which 80–90% falls in May–October) and a mean annual temperature of 24.5 ˚C [26]. The most common soil type is the montane lateritic red or yellow earth [26]. In addition, the study area is highly habitat heterogeneity with rich species composition and complex structure, the dominant species are *Lauraceae*, *Fagaceae*, and *Rubiaceae*. Moreover, the average canopy height in this area is 28.0 m, the density of trees with DBH (diameter at breast height) above 5 cm and 10 cm could reach 170 species and 150 species per hectare, respectively [27,28].

### Soil sampling

Total six plots with each of 3 equal-sized subplots ($20 \times 20$ m = 400 m$^2$) set in, were selected in the Jianfengling Long-term Research Station of Tropical Forest Ecosystem (each location is approved by National Park Administration of Hainan Tropical Rainforest) (Table 1). On April 20, 2019, soil samples at the depth of 0–10 cm were randomly collected from the subplot. In each subplot, five soil cores of 10 cm diameter from the middle to the four corners were mixed to form a composite sample. 18 soil samples were collected in the 6 sites. We also collected two samples with the same sampling method in JFL1 site, and total twenty soil samples were collected. The soil samples were packed in bags in a constant cool temperature, and then transported into the laboratory and stored at 4 ˚C in a refrigerator. After a week of soil stabilization, soils were passed through a 2 mm diameter mesh and divided into two parts. The first part was air-dried, homogenized, and used for the analysis of soil properties and the other was stored in the 4 ˚C for the incubation experiment.

In addition, in order to study the relationships between SOC (soil organic carbon) and NM-CH$_4$ emissions, the other twenty soil samples were collected from tropical rubber forest (RF1) in June, 2019 (Table 1).

### Soil chemical analysis

SOC was determined by concentrated sulfuric acid-potassium dichromate oxidation method [29]. Soil total nitrogen (TN) and soil total phosphorus (TP) were extracted by semimicro kelvin method and determined by automatic flow analyzer (PROXIMA 1022/1/1, ALLIANCE

**Table 1. Geographical information of soil sampling sites in the tropical rainforest.**

| Sites | Latitude (N) | Longitude (E) | Altitude (m) | Slope (°) | Aspect |
|-------|-------------|---------------|--------------|-----------|--------|
| JFL1 | 18˚43′54.74″ | 108˚53′10.04″ | 872 | 2.9 | north |
| JFL2 | 18˚43′53.23″ | 108˚53′17.13″ | 855 | 10.8 | northeast |
| JFL3 | 18˚43′53.20″ | 108˚53′22.90″ | 884 | 7.8 | north |
| JFL4 | 18˚43′55.71″ | 108˚53′30.93″ | 810 | 2.0 | northeast |
| JFL5 | 18˚43′54.05″ | 108˚53′30.23″ | 892 | 7.6 | north |
| JFL6 | 18˚43′59.70″ | 108˚53′39.10″ | 830 | 4.5 | southwest |
| RF1 | 19˚01′12.01″ | 109˚58′11.99″ | 127 | 1.0 | - |

**Table 2. Descriptive statistics of the soil properties in the tropical rainforest of Jianfengling National Nature Reserve.**

| Variable | Max | Min | Average | STDEV[e] | CV[f] |
|---|---|---|---|---|---|
| SOC[a] (g/kg) | 42.44 | 31.23 | 36.18 | 5.59 | 0.151 |
| TN[b] (g/kg) | 1.66 | 0.83 | 1.33 | 0.29 | 0.066 |
| TP[c] (g/kg) | 0.14 | 0.06 | 0.10 | 0.03 | 0.050 |
| SWC[d] (%) | 23.00 | 13.85 | 18.75 | 3.18 | 0.025 |
| pH value | 6.10 | 5.11 | 5.46 | 0.26 | 0.015 |

[a] soil organic carbon.

[b] Soil total nitrogen.

[c] soil total phosphorus.

[d] soil water content.

[e] standard deviation.

[f] coefficient of variation.

instruments, France) [30]. Soil water content (SWC) was determined by drying method. Soil pH value was determined by potentiometric method (the soil water ratio = 1:2.5), the basic physical and chemical indexes of soil were showed in Table 2.

## Incubation experiment

Autoclaving is the most widely used sterilization method [10]. We firstly sterilized the soil with high-pressure steam (30 min, 121 ˚C) to eliminate the influence of microbial $CH_4$, then freeze-drying and homogenization the soil. Finally, about 10 g of soils were transferred to a 100 mL serum bottle, which was then sealed with a high temperature resistant butyl rubber stopper.

Before others experiments, the soil NM-$CH_4$ emission characteristics at first 192 h (air samples collected at 0, 1, 2, 3, 4, 5, 6, 7, 8, 12, 24, 48, 72, 96, 120, 144, 168 and 192 h respectively) were studied from the incubation experiment in aerobic and anaerobic environment at 70 ˚C with natural soil water contents. The incubation treatments are as follows, different temperature, various soil water contents, hydrogen peroxide, and SOC contents.

1. To study effects of different temperature on NM-$CH_4$ emissions, the incubation experiments with natural soil water contents were conducted at 30 ˚C, 40 ˚C, 50 ˚C, 60 ˚C and 70 ˚C. The different incubation temperatures were achieved through incubators, and the anaerobic environment was created by blowing high-purity nitrogen [31].

2. According to the natural water contents of the soil samples, ultrapure water or freeze-drying were applied to adjust the soil water content into 8 gradients, which are 0%, 5%, 10%, 30%, 50%, 70%, 100%, and 200%, and the incubation was in aerobic environment at 70 ˚C for 24 hours. Because the incubation experiments with different temperatures showed that 70 ˚C was most beneficial to soil NM-$CH_4$ emissions.

3. The mass concentration of $H_2O_2$ concentrations were setting at 0%, 0.1%, 0.25%, 0.5%, 1%, and 2%. The incubation experiment was conducted in aerobic environment at 30 ˚C (with natural soil water contents) for 24 hours. Because the incubation experiments with different temperatures showed that the NM-$CH_4$ emissions at 30 ˚C are weak, which can avoid the influences of temperature on NM-$CH_4$ emissions.

4. Besides the twenty soil samples from the Jianfenling tropical rainforest, the other 20 soil samples were collected from the tropical rubber forest (Table 1). The NM-$CH_4$ emission

characteristics for the forty soil samples were measured by incubation experiment in aerobic environment at 70 ˚C (with natural soil water contents) for 24 hours.

## $CH_4$ concentration measurement

Air samples sampling was performed before and after incubation. Before sampling, syringe was used to blow several times to mix the gas and then took 1 ml gas sample. This operation was carried out to minimize the interference to the sample in the incubation flask. For the continuous observation experiment, an equal volume of compressed air or high-purity nitrogen needs to be injected into the incubation flask right after the extraction of gas sample to maintain the pressure inside the incubation flask.

The $CH_4$ concentrations in air samples were measured by a gas chromatograph (7890A, Agilent Co., USA) [26]. The gas chromatograph was equipped with a flame ionization detector (FID). The operating temperature was 250 ˚C and oven temperature was maintained at 90 ˚C. The fuel gas was $H_2$ (40 mL min$^{-1}$) and the combustion supporting gas was air (400 mL min$^{-1}$). The standard gas concentration of $CH_4$ was 2.0 ppmv, provided by Beijing AP-BAIF Gases Industry Co., Ltd.

## Data analysis

The emission rates of $CH_4$ was calculated based on the change of $CH_4$ concentration inside the incubation flask [14]. The total $CH_4$ emission fluxes for different SOC contents samples in 229 h incubation were calculated by fluxes in first 24 h divided the proportions of it to total fluxes. One-way ANOVA followed by Tukey's multiple comparison tests was used to establish significant differences of soil NM-$CH_4$ emission among the temperature gradient, the concentration of $H_2O_2$ solution, soil water content. The difference at $P<0.05$ level considered to be statistically significant. The statistical analyses were carried out in SPSS 22.0. Figures were generated using the software Origin 2018.

## Results

### Soil NM-$CH_4$ emission characteristics as incubation time increasing

The concentration of soil NM-$CH_4$ emission was measured at different incubation time, and results indicated that the emission flux of soil NM-$CH_4$ increased with incubation time in the first 129 h (Fig 1). Within 24 h, the NM-$CH_4$ emission flux showed a significant positive linear correlation with the time change ($P<0.01$ $R^2 = 0.98$). However, within 192 h, the NM-$CH_4$ emission rate showed a gradual decrease trend. In addition, the relationship between NM-$CH_4$ emission rate and incubation time was explained by the logarithmic function y = -0.19ln(x-10.19) + 0.11 ($R^2 = 0.97$) (Fig 2), which indicated that the soil would no longer release NM-$CH_4$ when the incubation time was up to 229 h. The soil NM-$CH_4$ emission at first 24 h account for 33.14% of total emission amounts in aerobic environment at 70 ˚C with natural soil water contents, which means that substrates related to NM-$CH_4$ emission during this period are sufficient. Therefore, it is important to study NM-$CH_4$ emissions within 24 hours.

### Influence of temperature on soil NM-$CH_4$ emission

To study the effect of temperature on soil NM-$CH_4$, soil samples were incubated at temperatures ranging from 30 to 70 ˚C with natural soil water contents under aerobic and anaerobic conditions, respectively. The results indicated that the $CH_4$ emission rate gradually increased ($+1.11\times10^{-3}$–0.15 µg/g SOC/h) with increasing of temperature at first 24 h (Fig 3). At lower

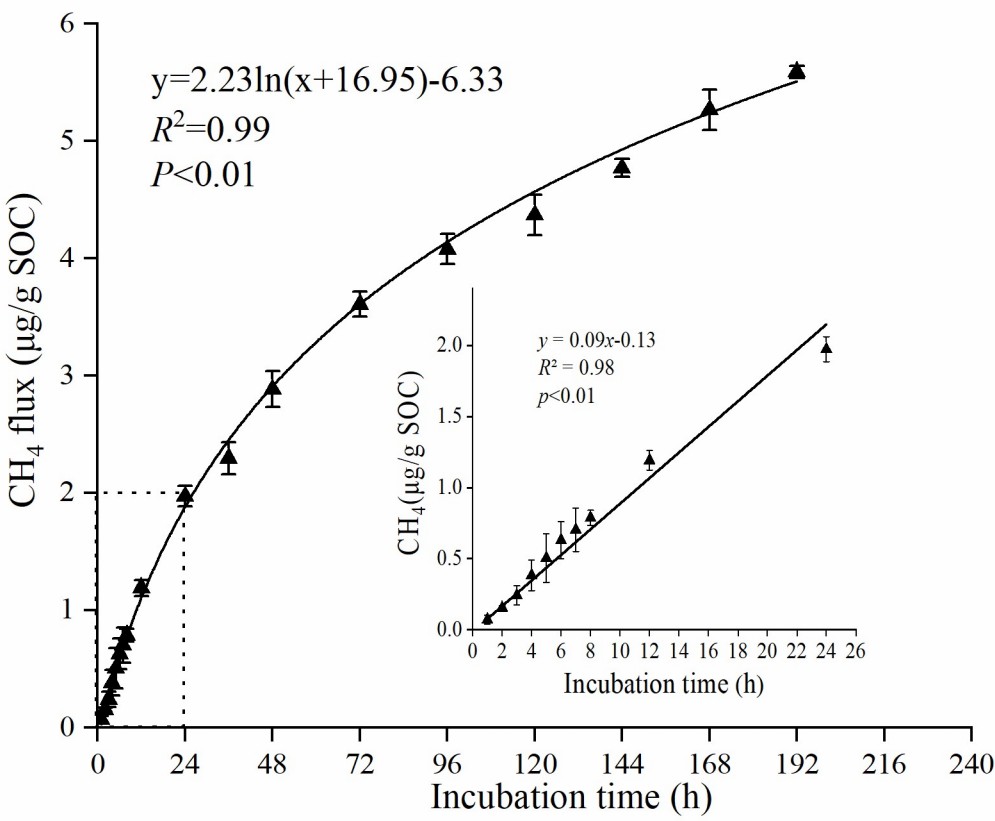

**Fig 1. Emission flux of soil NM-CH$_4$ in aerobic environment at 70 ˚C with natural soil water contents in first 192 h.**

temperatures (≤50 ˚C) conditions, the CH$_4$ emission rate was less sensitive to temperature change, and no significantly change was observed in the CH$_4$ emission rate, especially in the anaerobic environment (the emission rate of NM-CH$_4$ were lower than 0.06 μg/g SOC/h). However, the CH$_4$ emission rates changed dramatically when the temperature was high (60–70 ˚C) in both aerobic and anaerobic conditions, and the emission rates increased by 1–3 times per 10 ˚C (the emission rates of NM-CH$_4$ ranged from 0.05 to 0.15 μg/g SOC /h). In addition, there was a significant positive correlation between the emission rates of NM-CH$_4$ under aerobic and anaerobic conditions ($P<0.05$), and the emission rates under aerobic condition was always higher than that of under the anaerobic condition in a certain temperature.

### Influence of H$_2$O$_2$ on soil NM-CH$_4$ emissions

To study the relationships between soil NM-CH$_4$ emission and H$_2$O$_2$ contents, the soil samples were added with 5 ml H$_2$O$_2$ solution with different concentration (0%, 0.1%, 0.25%, 0.5%, 1% and 2%) at 30 ˚C with natural soil water contents. The results revealed that the NM-CH$_4$ emission rates increased with the increasing of H$_2$O$_2$ contents (+0.07–0.77 μg/g SOC/h), and the emission rates showed a significant positive linear correlation with H$_2$O$_2$ concentration ($P<0.01$, $R^2 = 0.96$) (Fig 4).

### Emission characteristics of soil NM-CH$_4$ at different soil water contents

The emission characteristics of soil NM-CH$_4$ in tropical rainforest at 70 ˚C were showed in Fig 5. The results of the study indicated that there was a mutation of NM-CH$_4$ emission rates

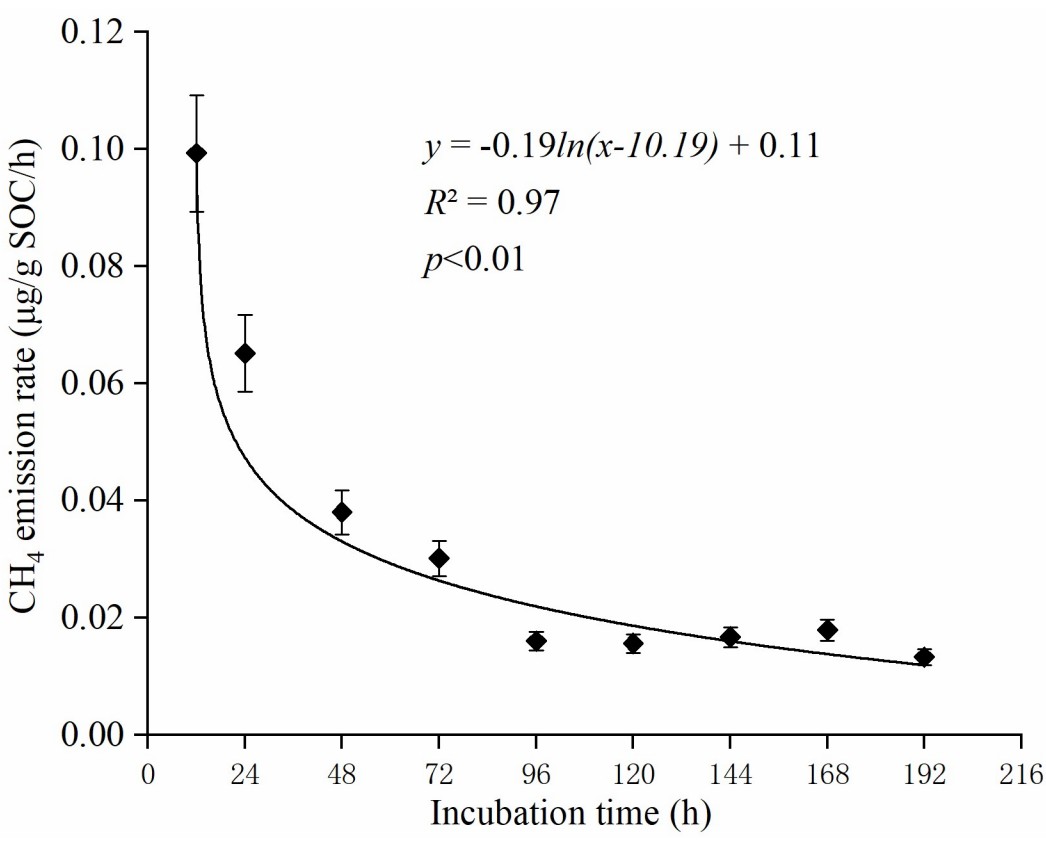

$$y = -0.19ln(x-10.19) + 0.11$$
$$R^2 = 0.97$$
$$p < 0.01$$

**Fig 2. Fitting relationship between NM-CH$_4$ emission rates and incubation time in aerobic environment at 70 ˚C with natural soil water contents in first 192 h.**

when soil water contents increased from 0% to 5%. The results also showed that soil water contents increasing promoted NM-CH$_4$ emissions (+0.02–0.31 µg/g SOC/h), and the effect decreased significantly at first and then decreased gradually. In addition, the CH$_4$ emission rate reached the peak at the soil water content of 5% (0.42 µg/g SOC/h). When the soil water content was 200%, the emission rate was 0.05µg/g SOC/h, which was lower than that of at soil water content of 0% (0.07 µg/g SOC/h). The results indicated that the excessive soil water contents inhibited the NM-CH4 emissions.

### Relationships between SOC contents and soil NM-CH$_4$ emissions

The soil NM-CH$_4$ emission fluxes were positively correlated with SOC in the both tropical rubber forest ($P<0.01$, $R^2 = 0.81$) and tropical rainforest ($P<0.01$, $R^2 = 0.79$), respectively. In addition, the total soil NM-CH$_4$ emission fluxes in the tropical rainforest (0.21±0.05 ug/g(dw)) were higher than those of in the tropical rubber forest (0.06±0.02 ug/g(dw)) (Fig 6). The average NM-CH$_4$ emission fluxes were 6.91 µg per gram organic carbon in the tropical forest of Hainan.

### Discussion

### Soil NM-CH$_4$ emission characteristics at different incubation time

There was a linear relationship between soil NM-CH$_4$ emissions and incubation time at the first 24 h (Fig 1), which was consistent with the results of Hurkuck et al. [24]. Our results also

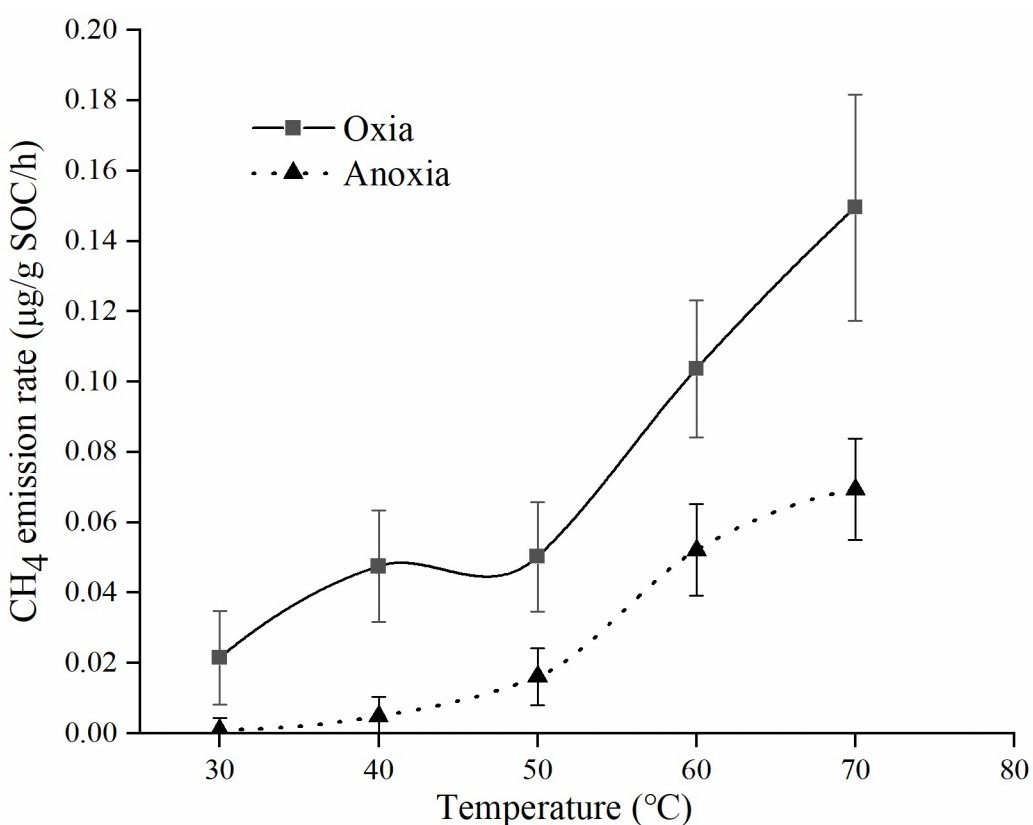

**Fig 3. The emission rates of NM-CH$_4$ at different incubation temperature from 30˚C to 70 ˚C and under aerobic and anaerobic conditions.**

showed that soil NM-CH$_4$ emission rates gradually decreased as the incubation time increasing from 24 h to 192 h. The NM-CH$_4$ emission rates were determined by the type and contents of NM-CH$_4$ precursors in soil [4]. It could be deduced that some of the precursors are consumed as the incubation time increased, resulting in a gradual decrease in the NM-CH$_4$ emission rates after 24 h in this study. Moreover, it could be concluded that there was no soil NM-CH$_4$ emission after 229 hours incubation from the logarithmic function (Fig 2), which may due to the contents of NM-CH$_4$ precursors were nearly exhausted. The emission rates of soil NM-CH$_4$ at first 24 h were higher in aerobic condition than those of in anaerobic conditions (Fig 3). Wang et al. [10] also found that the NM-CH$_4$ emission rates under aerobic condition was higher than that under anaerobic condition, and the differences intensified with the increase of temperature. Therefore, the following discussion focused on the influences of different temperature, H$_2$O$_2$ concentration, soil water contents and SOC contents on NM-CH$_4$ emission rates at first 24 h under aerobic incubation.

## Higher temperature, H$_2$O$_2$ concentration, soil water content promoted soil NM-CH$_4$ emission

The thermal degradation mechanism as a major mechanism of NM-CH$_4$ production is gradually accepted [4]. Previous studies reported that there were many functional groups including methyl, methoxy and methyl sulfide (NM-CH$_4$ precursors) existed in soil and they could produce CH$_4$ [18,32]. The NM-CH$_4$ emission rates increased gradually with the increasing of

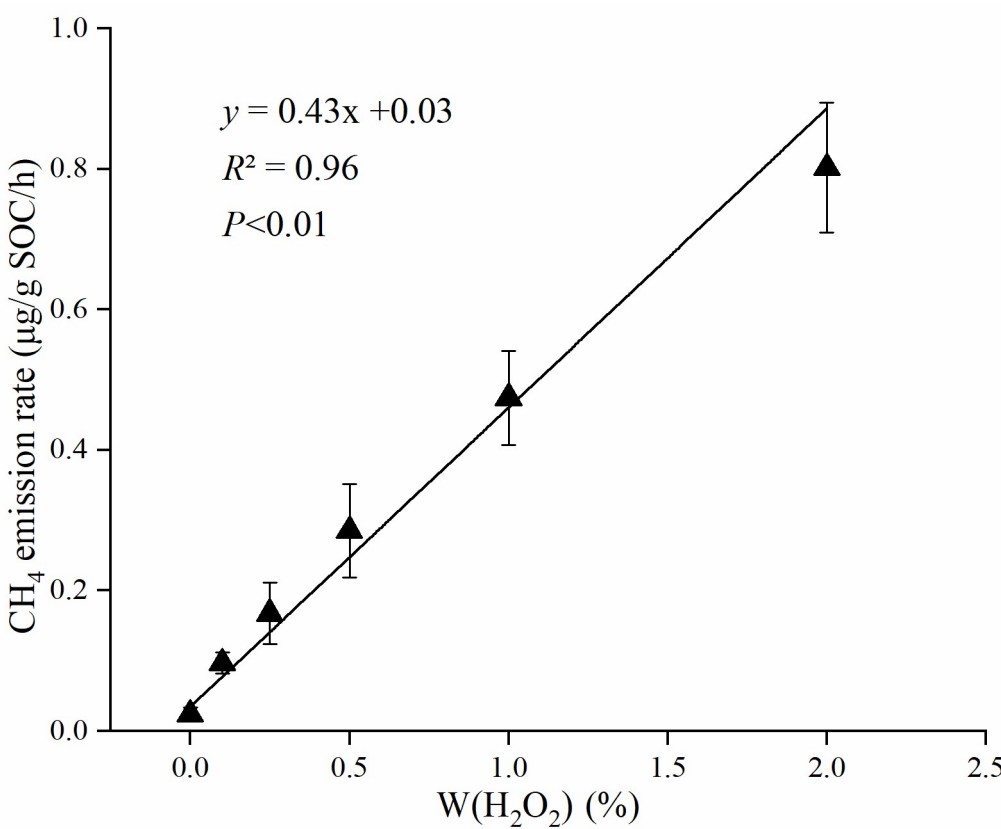

**Fig 4. The emission rates of NM-CH$_4$ at different concentrations of H$_2$O$_2$ (0, 0.1%, 0.25%, 0.5%, 1%, 2%) in aerobic environment at 30 ˚C.**

incubation temperature at first 24 h in this study (Fig 3). It indicated that high temperature could accelerate the NM-CH$_4$ emission, because the higher temperature could dissociate much more soil chemical bonds of NM-CH$_4$ precursors [10,23,25], and resulting in higher CH$_4$ production and emission rates.

Reactive oxidative species (ROSs, such as hydroxyl radical) are highly oxidative [33], capable of cleaving functional groups that can produce NM-CH$_4$. Soil biological activities can exude ROSs and soil itself can also produce ROSs through photochemical reactions with certain mineral oxides [34,35]. The results in this study showed that the soil NM-CH$_4$ emission rates were positively correlated with the H$_2$O$_2$ concentration at the first 24 h (Fig 4). The results of Jugold et al. [25] also found a similar pattern in aerobic soil. However, these results were different from the result of Wang et al. [10] that there was a logarithmic growth relationship between CH$_4$ emission rates and H$_2$O$_2$ concentration. The interesting thing is, some researchers have found the different patterns in plants, for example, Wang et al. [31] found that the plant pectin showed a logarithmic emission characteristic, while the dry leaves showed a linear emission characteristic. Those indicated that the carbon sources may be an important factor affecting CH$_4$ emission under the different H$_2$O$_2$ concentrations environment.

It is important to study the effect of soil water content on the release of NM-CH$_4$, because soil water content always changed in forest due to precipitation and evaporation. In this study, NM-CH$_4$ emission rates had a sudden increased as soil water content increasing from 5% to 10%, and it gradually decreased with soil water content increased from 10% to 100% at the

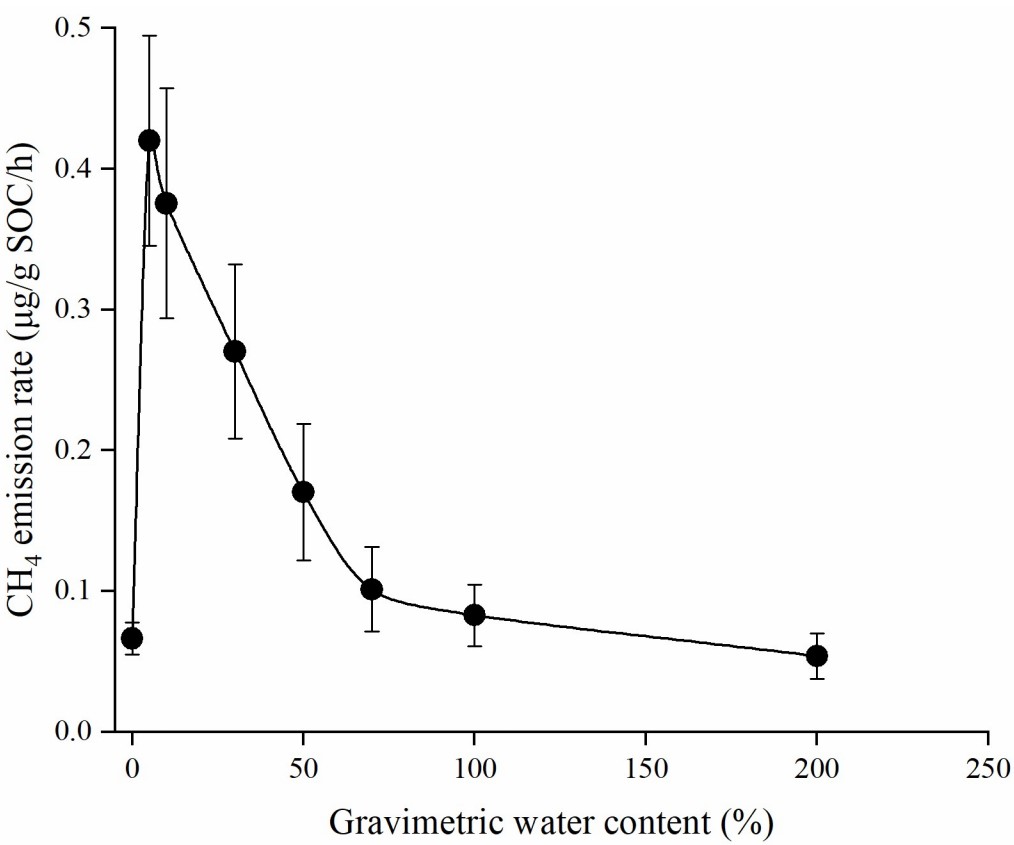

**Fig 5. The emission rates of NM-CH$_4$ at different soil water contents (0%, 5%, 10%, 30%, 50%, 100%, 200%) treatments in aerobic environment at 70 ˚C.**

first 24 h (Fig 5), which indicated that appropriate water can effectively promote NM-CH$_4$ emissions. However, CH$_4$ emission rate was decreased when soil waters content reached 200%, indicating the excessive soil water would inhibit NM-CH$_4$ emissions. The results are consistent with the previous results from Wang et al. [10] and Jugold et al. [25]. Soil water contents could affect NM-CH$_4$ release from chemical degradation of organic compounds, and it also affects oxygen concentration and the microenvironment where methanogenesis takes place [24]. More research is needed on the mechanisms of soil water contents affecting NM-CH$_4$ emissions in the future. For example, what is the internal mechanism of moisture affecting soil NM-CH$_4$ emission, how to explain its internal mechanism at the level of organic chemistry, and where is the critical point of the impact of moisture on soil NM-CH$_4$ emission?

## Potential of NM-CH$_4$ emission

This study found that the soil NM-CH$_4$ emission fluxes were significantly positively correlated with SOC content (Fig 6). When conducting experiments on NM-CH$_4$ emission from aggregates, Wang et al. found that there is a significant positive correlation between the NM-CH$_4$ emission rate of soil aggregates and the concentration of organic carbon, indicating that the role of the amount of organic carbon can determine soil NM-CH$_4$ emissions [10]. Similarly, the studies of Gu et al. and Hurkuck et al. also reported that there is a significant positive correlation between the release rate of soil NM-CH$_4$ and the soil organic carbon content [23,24]. It supported that soil NM-CH$_4$ was derived from soil organic matter. But what needs to be

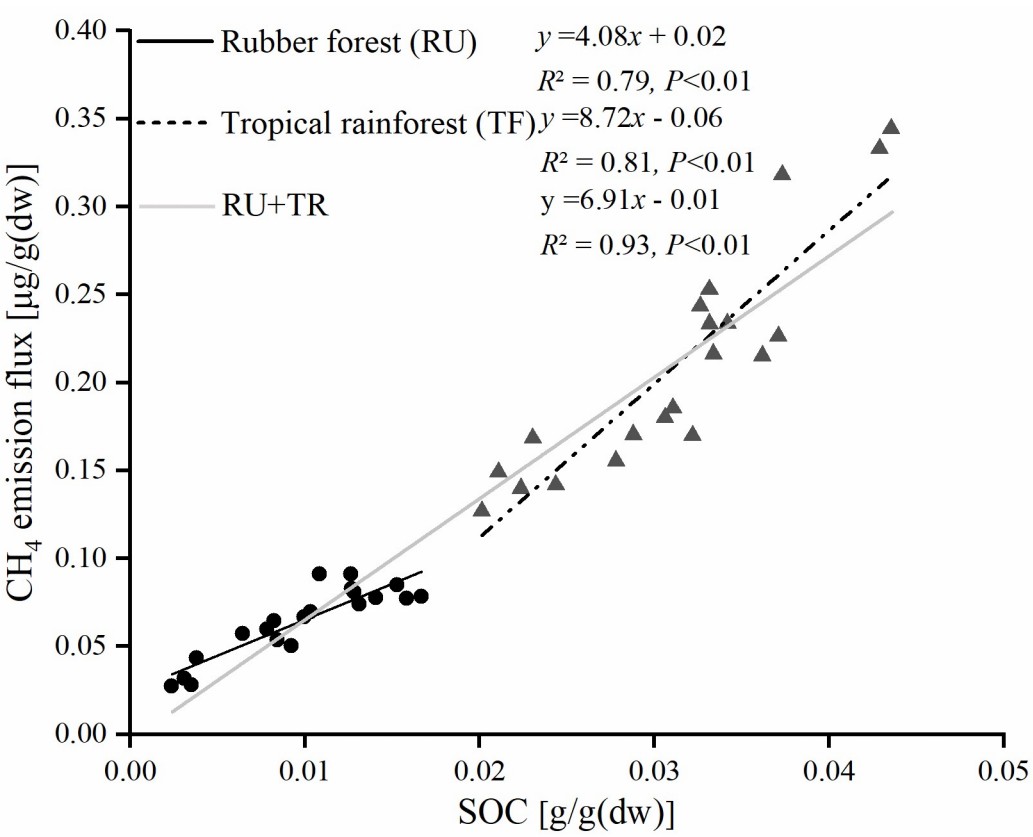

**Fig 6. The relationships between soil NM-CH$_4$ emission fluxes and SOC in the tropical rainforest of Hainan.**

further clarified is which organic molecules or structures in the soil are involved in the process of soil NM-CH$_4$ release. In the study of plant materials, it is found that the methoxy groups in pectin and lignin can be used as precursors of NM-CH$_4$ production [18,20], which provides ideas for the study of soil NM-CH$_4$ sources. Mao et al. used solid-state nuclear magnetic resonance technology to study soil humic acid, and found it contains the methyl-containing functional groups such as alkoxy, alkyl and alkyl groups [36]. This indicated that the humic acid was one of the precursors of soil NM-CH$_4$ emissions. How large is soil NM-CH$_4$ emissions in tropical forest? Our results showed that the average NM-CH$_4$ emission fluxes were 6.91 μg per gram organic carbon in the tropical forest of Hainan. This study first reported the emission potential of soil NM-CH$_4$, which could provide basic data for understanding the production mechanisms and potential of soil NM-CH$_4$ in the tropical rainforest. In fact, the production of NM-CH$_4$ in the natural environment is the result of combined different factors (such as temperature, oxygen, soil water, etc.).

## Conclusion

The NM-CH$_4$ could be produced in tropical rainforest soils under some condition. The NM-CH$_4$ emission would last about 229 h, and it exhibits a linear increase at first 24 h, but the increase rates decreased gradually as the incubation time increasing. The emission rates of NM-CH$_4$ under aerobic condition was higher than that under anaerobic condition at high temperature environment. The high temperature and H$_2$O$_2$ concentrations could significantly promote the emission of NM-CH$_4$ in tropical rainforest soil. The increasing of soil water

contents from 0 to 100% could promote the NM-CH$_4$ emission, while the excessive high soil water content inhibited the NM-CH$_4$ emission. There was significant positive correlation between SOC and NM-CH$_4$ emissions flux, which indicated that a great potential for NM-CH$_4$ emissions in tropical rainforest soils with high SOC.

In-depth research on NM-CH$_4$ plays an important role in further understanding the emission mechanism of soil CH$_4$ and more accurate prediction of climate change. However, the current researches on soil NM-CH$_4$ mainly concentrated on indoor incubation. How to conduct field in-situ observation (For example, two gaseous chemical reagents, monochloromethane and difluoromethane, are often used as inhibitors of CH$_4$ producing bacteria and CH$_4$ oxidizing bacteria, respectively. Can the two gases be directly used in the field and to achieve the purpose of inhibiting biomethane, thereby determining the release rate of NM-CH$_4$ under natural conditions), intuitively estimate its proportion to total CH$_4$, and explore more emission substrates and emission paths (For example, can the emission substrate and path be determined by the method of element labeling) still need to be further studied. This study found that temperature and moisture have a significant impact on NM-CH$_4$ emissions. The high-temperature and high-humidity soil environment of tropical forests, coupled with abundant organic carbon storage, can be considered as the best site for in-situ observation of microbial CH$_4$ and NM-CH$_4$. As we all know, model simulation is the best way to predict the current global change trend. If the process of NM-CH$_4$ is ignored, it may further increase the uncertainty of the regional and global carbon cycles predicted by current biogeochemical models. Therefore, based on the research that can distinguish between microbial CH$_4$ and NM-CH$_4$, algorithms for the production and emission of NM-CH$_4$ can be developed, verified, and integrated into the land surface model to better understand and predict large temporal and spatial changes.

## Acknowledgments

We are grateful for the sampling assistance given by the relevant personnel of Jianfengling National Nature Reserve, Hainan, China.

## Author Contributions

**Conceptualization:** Gaohui Jia, Wenjie Liu.

**Data curation:** Gaohui Jia, Huai Yang, Wenjie Liu, Han Mao, Tianyan Su.

**Formal analysis:** Gaohui Jia, Yamin Jiang, Wenjie Liu, Zhenghong Tan, Xu Wang.

**Funding acquisition:** Gaohui Jia, Wenjie Liu.

**Investigation:** Gaohui Jia, Wenjie Liu, Tingting Wu, Han Mao, Tianyan Su.

**Methodology:** Gaohui Jia, Qiu Yang, Huai Yang, Wenjie Liu, Zhenghong Tan, Xu Wang.

**Project administration:** Gaohui Jia, Wenjie Liu.

**Resources:** Gaohui Jia, Qiu Yang, Huai Yang, Wenjie Liu.

**Software:** Gaohui Jia, Wenjie Liu.

**Supervision:** Gaohui Jia, Wenjie Liu.

**Validation:** Gaohui Jia, Qiu Yang, Huai Yang, Yamin Jiang, Wenjie Liu, Tingting Wu, Han Mao, Tianyan Su, Zhenghong Tan, Xu Wang, Juelei Li.

**Visualization:** Gaohui Jia, Wenjie Liu, Juelei Li.

**Writing – original draft:** Gaohui Jia.

**Writing – review & editing:** Gaohui Jia, Qiu Yang, Huai Yang, Wenjie Liu, Tingting Wu, Han Mao, Tianyan Su, Zhenghong Tan, Xu Wang, Juelei Li.

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
