## [Decision Letter · Decision Letter 0]

26 Apr 2021

PONE-D-20-33065

Nonmicrobial methane emissions from tropical rainforest soils

PLOS ONE

Dear Dr. Liu,

Thank you for submitting your manuscript to PLOS ONE. After careful consideration, we feel that it has merit but does not fully meet PLOS ONE’s publication criteria as it currently stands. Therefore, we invite you to submit a revised version of the manuscript that addresses the points raised during the review process.

We look forward to receiving your revised manuscript.

Kind regards,

Dai-Viet N. Vo, Ph.D.

Academic Editor

PLOS ONE

Journal Requirements:

2. In your Methods section, please provide additional information regarding the permits you obtained to collect samples for the present study. Please ensure you have included the full name of the authority that approved the field site access and, if no permits were required, a brief statement explaining why.

Reviewers' comments:

Reviewer's Responses to Questions

**Comments to the Author**

1. Is the manuscript technically sound, and do the data support the conclusions?

Reviewer #1: Partly

Reviewer #2: Yes

2. Has the statistical analysis been performed appropriately and rigorously? 

Reviewer #1: Yes

Reviewer #2: No

3. Have the authors made all data underlying the findings in their manuscript fully available?

Reviewer #1: Yes

Reviewer #2: Yes

4. Is the manuscript presented in an intelligible fashion and written in standard English?

Reviewer #1: Yes

Reviewer #2: No

5. Review Comments to the Author

Reviewer #1: This incubation experimental study shows variations of NM-CH4 emission from tropical rainforest soils under different environmental conditions. The results confirm, as conclusively stated by the authors, the production mechanism of NM-CH4 emission by thermal degradation and reactive oxygen species oxidation. However, I’m expecting new findings beyond the results in previous studies except for the blank filling with new soil samples of rainforests in this study. Some other concerns are of the experimental treatments in the incubation.

1. Before other experiments, the soil NM-CH4 emission characteristics in the first 192h were studied from incubation experiments in aerobic and anaerobic environments at 70 °C with natural soil water content (Line 127-129). The temperatures up to 70 °C are not naturally realistic in forest soils, please explain your rationale of making the experimental temperatures (Line 133) beyond natural realities.

2. The anaerobic environment was created by blowing high-purity nitrogen in the experiment, the same as in Wang et al. (2011b) (Line134-135). In the study of Wang et al (2011b), they treated plant tissues and the anaerobic condition can be reasonably created and maintained by blowing high purity nitrogen. But in soil samples, there are tiny pores that trap oxygen in soil aggregations. I wonder the efficiency of erasing the oxygen in soil samples by merely blowing high-purity nitrogen. Some supplementary treatments are needed to ensure or check the anaerobic condition after blowing nitrogen.

3. What’s the situation of 200% soil water content (Line 138)?

4. The experimental results showed that the soil would no longer release NM-CH4 when the incubation time was 229 h (Line 180). The degradation of soil organic matters, which is the source of NM-CH4 production and emission, is a long term process up to hundreds to thousands years. Without mechanical explanation of the experimental phenomena, it’s hard to believe that the long term continuous degradation of SOM produces no NM-CH4 after 229h.

5. Fig. 5 shows that the NM-CH4 emission increased when soil water contents rose from 0% to 5%, and thereafter, more soil water content inhibited significantly the NM-CH4 emission. It seems that water molecules play key roles in the process of NM-CH4 production in soils. Could the authors explain it in the context of organic chemistry?

6. The total soil NM-CH4 emission fluxes in the tropical rainforest were higher than those in the tropical rubber forest (Line 232-234). This implies that the types of ecosystems (may also be the chemical composition of SOM in ecosystems) are key factors in NM-CH4 emission. I recommend some discussion of the results in this study to the published literatures (e.g. Wang et al., 2013).

Reviewer #2: Manuscript ID: PONE-D-20-33065

Title: Nonmicrobial methane emissions from tropical rainforest soils

Recommendation: Major revision

Comments:

Title: The title seems not complete, need to be more specific to your research.

Keywords: add one keyword: “tropical rainforest soils”

Abstract

• State the aim and the novelty of the study.

Materials and methods

• Citations needed. Please cite the papers of where the methods are adapted from.

Incubation experiment:

• Why choose the first 192h for incubation experiments?

• Why choose these 8 gradients, 0%, 5%, 10%, 30%, 50%, 70%, 100%, and 200%? Same question for “The mass concentrations of H2O2 were set at 0%, 0.1%, 0.25%, 0.5%, 1%, and 2% “

• Line 146, (table 1), should be Table 1.

CH4 concentration measurements:

• Does this method part refer to any reference?

Results and discussions

• Line 285-286: “More research is needed on the mechanisms of soil water content affecting NM-CH4 emissions in the future.” Please clarify which specific research can be done, give your suggestions.

• Line 298-301: “How to conduct field in situ observations, intuitively estimate its proportion to total CH4 emissions, and explore more emission substrates and emission paths still needs to be further studied.”Can you give some solutions in response to this problem? E.g. What methods can be conducted to solve this problem?

Conclusion

• Please state the importance of the study and how it can contribute to the research community.

• Please include the detailed limitations and what can be done for the future studies

6. PLOS authors have the option to publish the peer review history of their article (what does this mean?). If published, this will include your full peer review and any attached files.

Reviewer #1: No

Reviewer #2: No

---

## [Decision Letter · Decision Letter 1]

23 Jul 2021

Non-microbial methane emissions from tropical rainforest soils under different conditions

PONE-D-20-33065R1

Dear Dr. Liu,

We’re pleased to inform you that your manuscript has been judged scientifically suitable for publication and will be formally accepted for publication once it meets all outstanding technical requirements.

Kind regards,

Dai-Viet N. Vo, Ph.D.

Academic Editor

PLOS ONE

Additional Editor Comments (optional):

The paper has been properly revised. Thus, it can be considered for publication.

Reviewers' comments:

Reviewer's Responses to Questions

**Comments to the Author**

1. If the authors have adequately addressed your comments raised in a previous round of review and you feel that this manuscript is now acceptable for publication, you may indicate that here to bypass the “Comments to the Author” section, enter your conflict of interest statement in the “Confidential to Editor” section, and submit your "Accept" recommendation.

Reviewer #1: All comments have been addressed

Reviewer #2: All comments have been addressed

2. Is the manuscript technically sound, and do the data support the conclusions?

Reviewer #1: Yes

Reviewer #2: Yes

3. Has the statistical analysis been performed appropriately and rigorously? 

Reviewer #1: Yes

Reviewer #2: Yes

4. Have the authors made all data underlying the findings in their manuscript fully available?

Reviewer #1: Yes

Reviewer #2: Yes

5. Is the manuscript presented in an intelligible fashion and written in standard English?

Reviewer #1: Yes

Reviewer #2: Yes

6. Review Comments to the Author

Reviewer #1: The authors response to all comments raised by the reviewers. I must admit the complexity in the NM-CH4 emission procedures and recommend acceptance of the MS for publication in view that the methods and results of the study are sound in spite that it's hard to fully understand the mechanism of some phenomena of the incubation.

Reviewer #2: The authors have adequately addressed all of the comments. The manuscript can be accepted for publication

7. PLOS authors have the option to publish the peer review history of their article (what does this mean?). If published, this will include your full peer review and any attached files.

Reviewer #1: No

Reviewer #2: No

---

## [Editor Report · Acceptance letter]

28 Jul 2021

PONE-D-20-33065R1 

Non-microbial methane emissions from tropical rainforest soils under different conditions 

Dear Dr. Liu:

I'm pleased to inform you that your manuscript has been deemed suitable for publication in PLOS ONE. Congratulations! Your manuscript is now with our production department. 

Kind regards, 

on behalf of

Dr. Dai-Viet N. Vo 

Academic Editor

PLOS ONE